# Current Trends in the Development and Use of Personalized Implants: Engineering Concepts and Regulation Perspectives for the Contemporary Oral and Maxillofacial Surgeon

Alessandro Tel [1], Alessandra Bordon [2], Marco Sortino [2], Giovanni Totis [2], Lorenzo Fedrizzi [3], Elisabetta Ocello [4], Salvatore Sembronio [1] and Massimo Robiony [1,*]

1   Department of Maxillofacial Surgery, University of Udine, 33100 Udine, Italy;
    alessandro.tel@icloud.com (A.T.); info@sembroniomaxillo.com (S.S.)
2   Laboratory for Advanced Mechatronics, Department of Technology, University of Udine, 33100 Udine, Italy;
    alessandra.bordon@uniud.it (A.B.); marco.sortino@uniud.it (M.S.); giovanni.totis@uniud.it (G.T.)
3   Materials Science, Department of Technology, University of Udine, 33100 Udine, Italy;
    lorenzo.fedrizzi@uniud.it
4   Management Engineering, Polytechnic Department of Engineering and Architecture, University of Udine,
    33100 Udine, Italy; elisabetta.ocello@uniud.it
*   Correspondence: massimo.robiony@uniud.it

**Abstract:** The recently adopted Medical Device Regulation (MDR) has finally entered into force on 26 May 2021. As innovation and especially the advent of customized prostheses has deeply modified many surgical procedures in our discipline, it is imperative for the contemporary surgeon to become aware of the impact that the MDR will have on many aspects, including the choice of the manufacturer, the evaluation of the devices, point-of-care 3D printing labs, and medical software. In this paper, the authors tried to identify the cultural gaps in clinical practice that the MDR is supposed to fill. To achieve this purpose, a task force of experts was reunited, including CMF surgeons with direct expertise in medical software and 3D printing, mechanical and material engineers, facing the topic of the MDR from a multidimensional perspective. In this article, surgeons and engineers review many crucial aspects concerning the points of the regulation that mostly affect the field of implantable devices for the cranio-maxillo-facial skeleton. The result of interdisciplinary research is a paper aiming to provide surgeons with the knowledge on the fundamental processes of additive manufacturing, increasing the clinician's awareness on the evaluation of a customized implant before surgery and on the underlying regulatory framework.

**Keywords:** printing; three-dimensional; maxillofacial prosthesis implantation; mechanical phenomena; maintenance and engineering; hospital; maxillofacial surgery

## 1. Introduction

As of 26 May 2021, the EU Medical Device Regulation (EU MDR) has come into force [1], replacing the EU Medical Device Directive 93/42/EEC [2,3].

This new set of rules has implications for both in-house facilities, namely, small laboratories which are integrated in hospitals and are managed partially or entirely by clinicians, as well as for large-scale companies, which only provide the final product to be implanted.

In-house laboratories are nowadays present at many institutions, but without a regulatory framework, processes are highly heterogeneous and poorly standardized. It is important that such small facilities implement what is called a Quality Management System (QMS), through which they can certify the quality of processes [4]. In modern hospitals, such laboratories generally represent the place where virtual surgical planning is performed by surgeons, or, in the most advanced realities, where also non-implantable devices, such as molds and surgical guides, are manufactured using 3D printers.

Implantable devices deserve a separate discussion since they are produced by external companies using additive manufacturing technologies. Implants are ordered and received ready for surgery, and oftentimes no validation of their quality is performed by surgeons, that generally have little or no knowledge on production processes related to cranio-maxillo-facial (CMF) implants fabrication.

While it is clear how to test and validate a stock device, considering the reproducibility of the production processes across all replicas, customized implants do not obviously show the same predictability and might therefore be more subject to errors. In recent years, reconstruction of complex defects of the facial skeleton, or replacement of a dysfunctional temporomandibular joint (TMJ) have increasingly relied on patient-specific implants (PSIs), also known as customized implants. Subsequently, a number of companies flourished across the globe, providing personalized implants for all types of defects. For instance, a review reports that in 2019 there were 27 different TMJ replacement systems, of which 21 are customized. Most importantly, of such devices, only four had been approved by regulatory bodies, two by the Food and Drug Administration (FDA) in the US and two by the Australian Register of Therapeutic Goods [5]. Intuitively, the number of such companies has continued to grow, and this occurred especially in Europe, where the apparent lack of a well-defined regulatory framework encouraged smaller companies to produce their own implantable devices [6,7]. Moreover, regulatory authorities of many non-EU countries have already posed—or might pose in the near future—similar restrictions, therefore this topic should deserve considerable attention from surgeons worldwide.

In addition, the new regulation will also affect point-of-care 3D printing laboratories that have continued to rise within hospitals, supported by the most recent literature [8], with the aim to make services of virtual planning and rapid prototyping immediately available to surgeons.

To analyze, address, and understand such issues, the CMF surgery department of the University Hospital of Udine has brought together a task force made of experts in maxillofacial surgery, medical modeling, additive manufacturing, production technologies, and material engineering. The aim of this paper is to illustrate the complexity of the topic concerning patient-specific implant production from a multidimensional point of view, especially in relation to manufacturing technologies and final product evaluation, with the hope to provide surgeons with the indispensable knowledge to responsibly evaluate a customized craniofacial implant before surgery, as well as to better define related regulatory aspects.

## 2. Engineering Principles for Clinicians

### 2.1. Additive Manufacturing Concepts for the CMF Surgeon

Metal implants for CMF surgery are generally produced using additive manufacturing (AM) technologies that allow to 3D print a lot of metal types and especially titanium and its alloys. 3D powder bed fusion (PBF) printing is a production process that consists of the progressive deposition along the vertical axis of ultra-thin layers of metallic powder, melted layer by layer using an energy source appropriately guided depending on the desired geometry. In the case of overhanging features in respect to the growth direction of the fusion, the geometry to be printed will often need to be supported by column-shape elements that have the double function to ensure a solid base for the protruding features and to dissipate heat [9].

The most important advantage of these technologies is to allow the creation of very peculiar and complicated shapes, that could not be produced with the classic subtractive machining, like milling. For this reason, 3D PBF printing is perfectly suitable to the challenges imposed by the prosthetic replacement of the facial skeleton, since it allows the creation of prostheses that perfectly replicate even the most complicated bone anatomies.

The printing process is very delicate, it takes place in controlled atmosphere and the conformity of the produced pieces depends firstly on a correct setting of the parameters of the machine and then on a good positioning of the piece in the melting chamber as well as

on an appropriate design. The most common technologies for 3D PBF printing are SLM (selective laser melting) and EBM (electron beam melting). The main difference between them is the energy source, a laser for the former and a high energy electron beam for the latter [10].

EBM vs. SLM

It is useful to understand the main properties of the technology and to be aware of its limits and criticalities while designing a piece with particular resistance, finishing or precision features to be produced with 3D printing, since SLM and EBM are different. The main differences between them are the following [9]:

- EBM technology has usually a higher building rate in respect to the SLM, therefore it is more suitable for the production of complex shapes, for example lattice structures; EBM technology has usually a higher building rate compared to SLM. This is due to two main factors: first, layer thickness has higher values (50–100 μm in respect to the 20–100 μm of the SLM process); secondly, the immediate electron beam motion from one location to another, thanks to the instantaneous response of magnetic coils, can considerably speed up component fabrication (for example, electron beam speed can reach 8000 m/s in ARCAM machines [11], with the laser reaching 7 m/s in EOS machines [12]). For instance, in the production of truss-like structures (lattice structures), EBM is to be chosen;
- The surface finishing of a component produced with EBM is much lower in respect to an SLM product. Is it sufficient to consider that the roughness parameters of EBM "as built" specimens are about twice those of SLM "as built" ones: in fact the roughness of EBM is 30–40 μm while for SLM specimens is 11–18 um. This is mainly due to the fact that the layer thickness, powder size, and melting pool size in the case of SLM are half the one used in EBM specimens [13];
- The components produced with SLM undergo strong thermic gradients during the 3D printing process, since the preheating temperature of the powders is generally low, since the temperature of the chamber in SLM process is the environment temperature, assumed to be 293 K [14]. That is why they need some thermic post treatments in order to reduce the residual stresses that take origin inside the material because of the rapid and iterative phase change of the metal (solid–liquid–solid). On the contrary, in the EBM technology the temperature of the powder bed is higher and ensures the absence of thermal stresses inside the melted material, around 870 K during the melting process [15];
- On the contrary, maintaining the powder bed at high temperature has a bad influence on the quality of the microstructure of the melted metal that results coarse with large grains. This fact has a direct influence on the mechanical characteristics of the material: an SLM component, in fact, has higher resistance to traction while being less ductile in respect to an EBM product [16,17].
- In the EBM, the pre-heating of the powder allows the unmelted particles to bind together and to act as a support for the overhanging geometrical features. Generally, EBM products need fewer physical supports in respect to the SLM ones.

The correct positioning of the supports, especially when using the SLM, accounts for the good quality of the final piece in terms of dimensional precision, surface finishing, and defects. The main functions of support structures are [18]:

- Withstand deformation or even collapse of processed material caused by gravity during the manufacturing process;
- Mitigate the effects of thermal gradients generated during production, since thermal distortions may lead to cracks, curling, sag, delamination, and shrinkage;
- Anchor the part to the build platform;
- In PBF, they stop any layer shifting during the re-coating phase.

In summary, the SLM process requires an accurate study for the positioning of the supports and a fundamental post processing phase, in which the component is cleaned from the residual powder, the supports are removed, and the piece is thermally treated to relieve residual stresses and avoid cracking. EBM products, as seen before, do not need such a long post processing, except for a very accurate powder removing phase. Of course, rough surfaces "as built" will have a higher roughness value and might need some finishing operations to enhance their quality.

Figure 1 schematically represents SLM and EBM processes and Figure 2 illustrates CMF implants prototyped using SLM and used for surgeries.

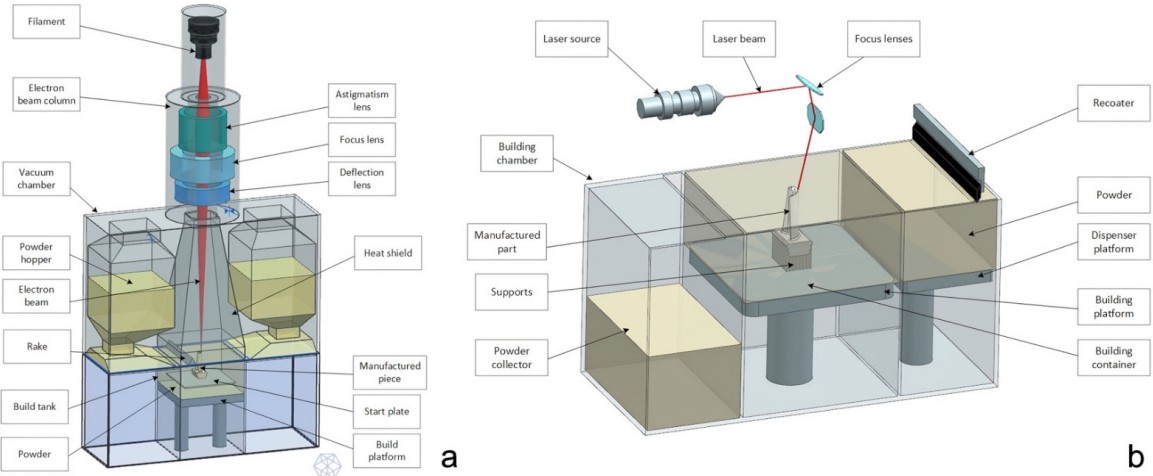

**Figure 1.** EBM and SLM technology principles: (**a**) EBM; (**b**) SLM.

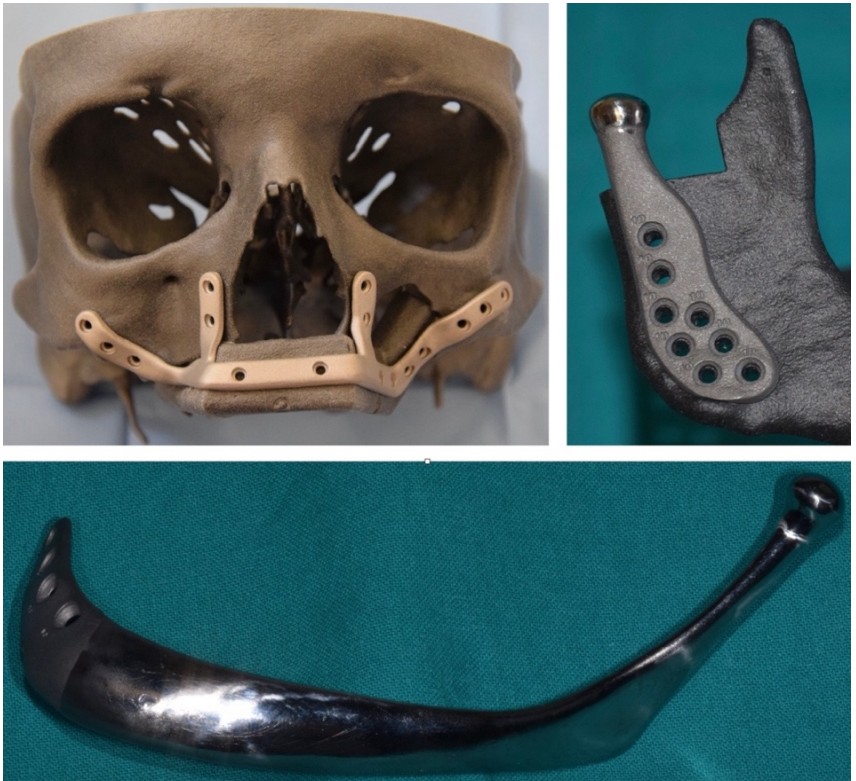

**Figure 2.** 3D printed implants manufactured by using SLM and used for surgeries at our institution include customized reconstruction plates, temporomandibular joint prostheses, and full mandibular prostheses used for reconstructive purposes in oncologic patients.

### 2.2. How to Evaluate the Requested CMF Implant? A Guide for the Surgeon

2.2.1. Principles of Mechanical Defects Formation

Before learning how to perform an effective visual inspection, it is necessary to understand some main defects connected with the 3D PBF printing. They can be divided into macroscopic and microscopic, and they usually depend on the wrong setting of the parameters of the machine, but they are even related with some criticalities of the geometry to be printed and its positioning inside the melting chamber. It is of prominent importance to underline that almost every defect can be avoided or corrected by conducting a systematic analysis and optimization of the parameters involved in the 3D printing process (Table 1).

**Table 1.** Criticalities that might occur during the manufacturing process and design recommendations to avoid them.

| Critical Feature Description | Recommendation for Prosthesis Design |
| --- | --- |
| Vertical thin walled structures with thickness from 1 to 0.1 mm | Avoid thin walled parts having thickness smaller than 0.2 mm |
| Pipe-like, hollow cylinders with different external diameters and thicknesses | |
| Small quarters of spherical shells presenting undercut surfaces | |
| Abrupt transition from semi-void section to full section | Avoid abrupt section variations whenever possible |
| Small full cross section parallelepipeds and cylinders having variable fillets/radii at their bases | It is possible to print small features with small or no fillets/radii at their conjunctions with other surfaces |
| Horizontal cylindrical holes with diameters in the range 2–8 mm, without internal supports | Horizontal holes are feasible having maximum diameter of about 8 mm, but they can be inaccurate; better results can be achieved by assuming a drop-like cross section shape |
| (Undercut) surfaces with different slope | Surface quality is affected by the staircase effect, but it can be partially improved by varying the orientation of the part with respect to the platform. Small undercut surfaces are feasible without supports |

- Porosity: the presence of pores inside the metallic structure could be critical for the fatigue resistance of the component [19]. Notably, porosities usually represent the trigger points for cracks propagation. These pores usually measure from 1 to 20 μm and can extend up to the surface.
- Balling: melt ball formation occurs when the molten material solidifies into spheres instead of solid layers. The result is a rough and bead-shaped surface that produces an irregular layer deposition with detrimental effects on the density and quality of the part [20]. Balling increases the surface roughness and contributes to the formation of a large number of pores.
- Surface defects: the presence of a rough and non-homogeneous surface represents a critical issue for the final component. In PBF processes, surface roughness has two main contributors: the stair-stepping effect due to the layer-wise production, and the actual roughness of the metal surface. The surface finishing depends on the surface orientation with respect to the growth direction [21]. In particular, downward and upward surfaces are known to have considerably different roughness properties. The former present much lower surface quality. It is important to highlight the dependence of the fatigue resistance on the surface roughness of the stressed surface: the higher the roughness the lower the fatigue performance of the component.

- Geometric defects: the PBF produced parts may exhibit different kinds of dimensional and geometric deviations from the nominal model: shrinkage and oversizing are the most common ones. Other sources of inaccuracy are represented by warping (a curling phenomenon that yields a curved profile of down facing surfaces intended to be flat) and by the formation of super elevated edges. These phenomena deteriorate the surface topology and the dimensional accuracy while interfering with the efficiency of the recoating system as well as damaging the adjacent pieces. Other distortions affect critical features like thin walls, overhanging surfaces, and acute corners [22].
- Residual stresses, cracking, and delamination: the SLM printing is known to create in the molten components great residual stresses that could result in cracking or delamination when the arisen tensile stress exceeds the ultimate tensile strength and overcomes the binding ability between two adjacent layers [23]. As a consequence, a partial disconnection of the part from the base plate could occur.
- Microstructural inhomogeneities and impurities: PBF processes involve highly localized high-heat inputs during very short beam-material interaction times that will therefore significantly affect the microstructure of the part [24], leading to the formation of microstructural inhomogeneities or nonequilibrium microstructures that could have a detrimental influence on the mechanical and functional performances of the part. These kinds of defects includes impurities (inclusions, contaminations from other materials, and formations of surface oxides), grain size characteristics, and crystallographic textures. Furthermore, the presence of unfused powders within pores or in the form of satellite powder clumps could represent a severe problem for the safety of the device [25].

### 2.2.2. Visual Inspection

While evaluating the conformity of components printed with 3D PBF technology, it is a good habit to keep in mind all of these defect types in order to perform an efficient visual inspection. This is especially true while evaluating a CMF component that is under assessment for surgical implantation. The leading idea is the smoother and more homogenous the surface, the better the quality of the component. It will be of key importance to evaluate the absence of isolated pores (1), detectable with bare eye, and of inhomogeneous accumulation of material, like clots (2), burrs or thin leaves of material (3), since they represent structural discontinuities, possible zones of stagnation of unmelted powders or of undesired fusion secondary products (10–150 μm). If not correctly removed, they could promote the proliferation of pathogens. Figure 3 refers to a mandibular prosthesis ordered by our institution and rejected because of unwanted defects. In this example, even a bare eye inspection, if performed with attention and according with the aforementioned explanations, reveals unacceptable imperfections.

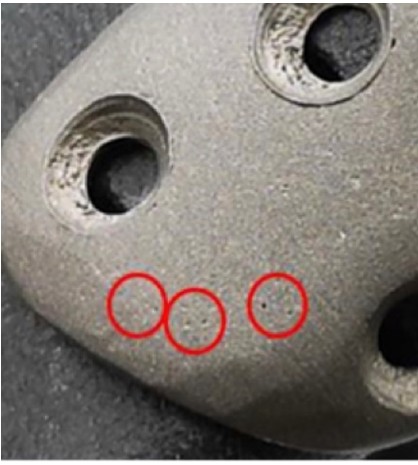 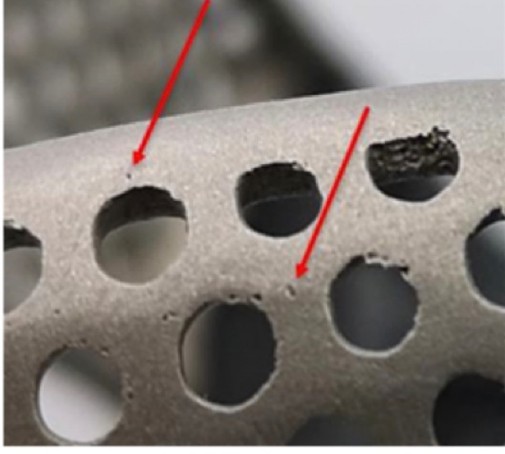

**Figure 3.** *Cont.*

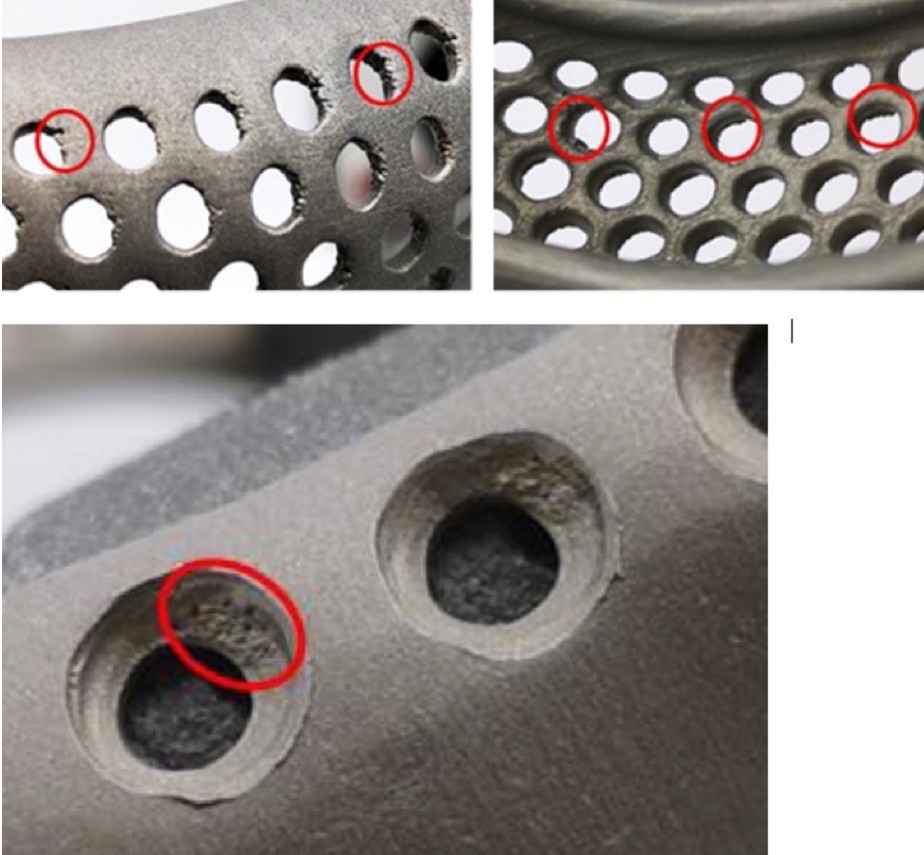

**Figure 3.** Example of accurate visual inspection conducted on a mandibular prosthesis ordered before MDR entered into force according to the explanations provided in Section 2.2.2. The manufacturer did not comply with requirements of the new MDR. In particular, undesired amounts of material can be seen on the edges of screw holes, appearing as plugs, clots, or thin leaves. Such imperfections are at risk of detachment during screw fixation and are freed into the body. Moreover, the whole implant surface is filled with small burrs, which are the ideal support for biofilm adhesion.

### 2.2.3. Laboratory Inspections

On the other hand, microporosity and microcracks are not detectable with a visual inspection, therefore special tests need to be carried out to observe them, as they could represent the trigger point for larger cracks especially in fatigue stressed components. Invasive and not invasive techniques could be used for this purpose. The former ones include:

- Metallographic analysis: the component is cut in different positions and the sections are analyzed with a scanning electron microscope, SEM;
- Penetrant liquid testing: the component is immersed in a fluorescent liquid with high capillarity and is then analyzed under a Wood lamp to see where the liquid propagated—this test highlights only surface porosities that could be critical for biological contamination. Penetrant liquid test is a non-destructive control method, nevertheless the analyzed part could undergo contamination by the fluorescent liquid [26];

whereas the latter ones include:

- Ultrasonic inspection [27,28];
- Industrial computerized tomography [29,30].

However, to be sure of the safety of the component, a biocompatibility panel of tests should be performed in order to understand if this kind of porosity is critical for contaminants and for bacteria proliferation.

*2.3. Design Tips to Improve the Final Result*

Porosity and geometrical defects are often the consequence of a mispositioning of the supports or of a wrong orientation of the part in the melting chamber. Therefore, while creating and designing the virtual model, it is fundamental to follow some guidelines for the 3D PBF printing, which need to respect some feasibility requirements to avoid possible issues during the production steps:

- Identify the functional surfaces, where good finishing and precision are essential, in order for the producer to avoid their down facing positioning and to avoid placing supports on it.
- Avoid inserting in the design undercut features, i.e., that have some surfaces overriding a critical angle of inclination in respect to the working plane. The critical angle depends on the process, on the material, and on the parameters of the machine, but generally it is known to be around 45°. Surfaces that override this inclination need to be supported and therefore could undergo a lack of quality [31].
- If not strictly necessary, avoid the positioning of holes with a non-parallel axis in respect to the growth direction of the piece. Do not use too long bridge elements. If it is possible, position holes, pits, and through holes with the axis as closer as possible to the vertical position in order to prevent the formation of unwanted material accumulations.
- Avoid too small holes and too thin features, that would be very affected by the residual stresses and would probably deform. If such details are necessary, they will be performed using a subtractive method once the part is printed.

**3. Implications for CMF Implants in Light of the New MDR**

Referring to the new MDR [1] in force from June 2021 it will be necessary to supply a "good manufacturing" certificate for the prosthetic device and all the proofs to demonstrate the avoidance of all the risks connected to every aspect of the productive process.

This new set of rules requires a validation of several aspects of the production and of the main features of the final device through a risk analysis and all the tests that are required. It will be necessary to give evidence of the mechanical resistance of the device, surface properties, biocompatibility (including the entire process and all of its contaminants) and sterility.

Such assessments require specific laboratory tests, but by learning how to control some significant features on the device, even a basic visual inspection might confirm the good manufacture of the prosthesis and could immediately highlight some criticalities.

The Annex I of the MDR imposes to fulfil the requirements regarding chemical, physical, and biological properties of the chosen materials and of all the substances involved in the production process [1]. Specifically, it is necessary to:

1. Give evidence of the absence of toxicity in the materials of which the prosthesis is made and in all the residuals of the contaminants as well.
2. Ensure the compatibility of the manufacturing materials and substances with the biological tissue, cells, and corporal fluids.
3. Ensure the compatibility among the different parts of a device that is intended to have several components to be implanted.
4. Verify the impact of the production process on the materials properties.
5. Study the mechanical properties of the materials focusing on the strength, ductility, resistance to cracks, resistance to wear and to fatigue.
6. Study the surface properties.
7. Confirm that the final device respects all the chemical and physical requirements.

Furthermore, devices shall be designed and manufactured in such a way as to reduce as far as possible the risks posed by substances or particles, including wear debris, degradation products, and processing residues that may be released from the device itself [32].

Devices and their manufacturing processes shall be designed in such a way as to eliminate or to reduce as far as possible the risk of infection preventing the microbial contamination.

These topics are of high importance when considering additive manufacturing products, taking into account the aforementioned issues about 3D printing techniques.

## 4. Possible Corrective Actions

In order to guarantee the safety of the devices and to protect the final user, it is recommended to plan the execution of specific tests and laboratory experiments. A complete test panel should include:

1.  An evaluation of the biocompatibility not only of the stock material, but especially of the finished product. This kind of analysis involves several tests summed up in the regulation ISO 10993:2021 and should be performed on the finished product. It is fundamental to trace every single phase of the production process in order to understand the various contaminants that enter in contact with the prosthesis. Once the test has given positive results for a specific device, its whole process is intended to be safe from the biocompatibility point of view. The result of the analysis is considered acceptable as long as anything in the process is altered, causing a modification in the contamination chain.
2.  A microbiological (bioburden) and sterility test according to the ISO 11737 to evaluate the efficiency of the sterilization and of the cleaning phase. Chemical characterization test according to the ISO 17025.
3.  Resistance tests on melted material samples: tensile test to evaluate the $\sigma$-$\varepsilon$ curve, rotating fatigue test to understand the performance of the material when stressed with an alternated symmetrical cycle.
4.  Finite element analysis (FEA) structural characterization of the final geometry of the prosthesis, according to a defined stressing conditions protocol that simulates the actual working conditions of the prosthesis, in order to investigate the static resistance and the efficiency of bone fixation. The magnitude of the loads applied should be then increased by a security factor (Figure 4).
5.  In case of articulations involving cycled loads of great intensity or of prosthesis in which critical geometrical features are stressed, a fatigue test on a final component should be arranged. This test should be performed with a setup representing the worst-case loading condition. Indeed, a dedicated ISO for CMF implants does not exist yet, but a proper setup should be created taking as guidelines the ISO 14801:2007 or ISO 16428:2005.
6.  Porosity analysis through metallographic cuts, ultrasound technology or magnetic resonance, to understand the homogeneity of the internal metallographic structure and to investigate the surface porosity as well.
7.  Wear test in aggressive environment replicating the actual working conditions of the prosthesis, together with a count of the residual particulate released by the contact surfaces after the test in order to give evidence of the absence or presence of fretting phenomena. In this case, ISO 17853:2011 should be taken as reference.

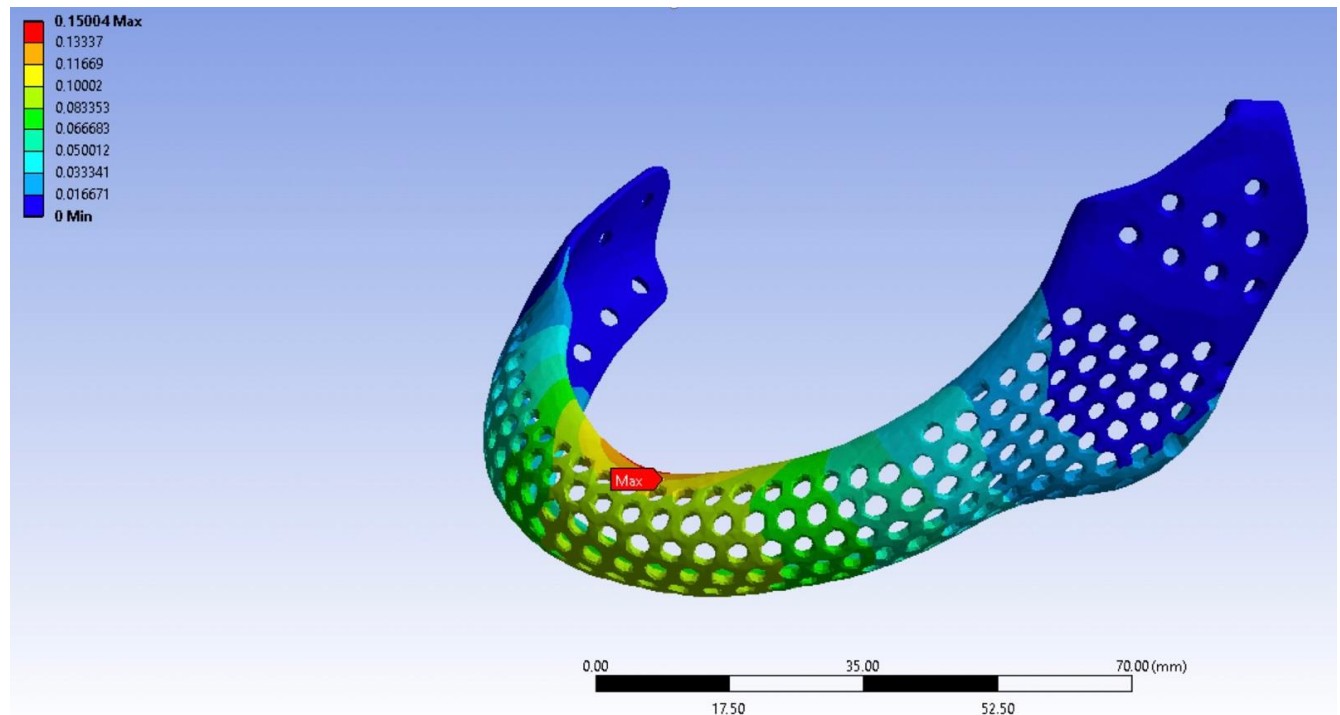

**Figure 4.** Finite element analysis performed on the final design reveals the total deflection of the component under the load of a cantilever force whose intensity is far beyond the physiological working conditions. The maximum calculated deflection together with the maximum stress analysis reveals if the component is suitable for implant or not.

## 5. Suggestions for Your Point-of-Care 3D Printing Lab

The entry into force of the new regulation also affects point-of-care (PoC) 3D printing laboratories, which are an emerging reality that has increasingly gained importance and diffusion within many hospital centers. The presence of a PoC 3D print lab has the advantage to consistently speed up the processes of virtual surgical planning and production of simple surgical non-implantable devices, including surgical guides, or anatomical models used for case studies. However, alike third-party companies, in-hospital PoC laboratories must adapt to the general requirements of using medically certified software and printing biocompatible materials.

Concerning the first point, the MDR has introduced important conceptual changes about software intended for clinical use, which needs CE marking, therefore many centers relying upon freeware and opensource software need to adopt medically-certified software to conduct virtual surgical planning for real patients [33].

In addition, considering that 3D printers, like software, are used to manufacture other medical devices, such as surgical guides and splints, 3D printers and the materials used should in turn be certified for a medical use. In particular, considering that SLA (stereolithography) 3D printers are the most widespread across hospitals, it is important that photopolymers used undergo biocompatibility tests to ensure that no toxic materials are released in the body.

Another fundamental point regards post-market surveillance systems, considering that all processes performed within a PoC lab, or a third-party company, must be traceable using univocal IDs. It is important that each 3D printed manufact is labeled with a univocal code, and that registers are arranged to keep track of virtual planning procedures and printed parts.

## 6. Discussion

The MDR will involve significant modifications in the field of CMF implants, which have become nowadays indispensable devices to perform accurate patient-specific reconstructions of the facial skeleton. However, with the previous legislation it was hard to establish whether CMF implants underwent all the necessary evaluations to confirm their safety for patients.

For this paper, mechanical, clinical, and material engineers mutually shared their knowledge with surgeons to suggest a sequence of analyses and tests that should be implemented for each aspect of the implant connected to hazard, including for instance the risk of rupture, cracking, particle release, deformation, biological damage.

Our idea is that surgeons should become acquainted with concepts related to the fabrication of the implant and should understand the key principles of mechanical defects related to 3D metal printing and how they originate, not only to perform a rudimental assessment of the implants they plan to use, but also to be more participative in studies involving the development of new devices. Current literature provides evidence that surgeons are progressively more interested in designing their own implants, or, at least, in taking active part in the process of developing novel devices for surgeries. In our opinion, the surgeon must strictly supervise and actively participate in the design of the implant, while engineers are supportive for technical aspects. In a multidisciplinary team, surgeons have the deeper anatomical knowledge and are aware of possibilities offered by surgical access that might limit the positioning of an implant, a factor the engineer hardly considers while evaluating a raw skeletal model. On the other hand, engineers have a deeper knowledge of CAD packages and are able to perform structural calculations and coupling design.

The MDR impacts on the reclassification of several devices, as certain CMF devices are subject to a step forward to a higher class. For instance, partial or total articular prostheses have been included in class III [1]. As mentioned, the number of companies providing CMF implants underwent an exponential growth in the last years, leading to a variety of devices, that might show consistent differences among each other. Moving a device to a higher risk class might result in a substantial increase in costs and conspicuous delays for the recertification process. This may discourage manufacturers from continuing to sell some of their devices, considering that many of them might not be compliant with the MDR after the class shift [34]. On the other side, customized CMF implants have become unrenounceable in modern CMF surgery, considering the significant reduction in surgical times and improvement in accuracy they allow to obtain, thereby representing a limitation for several medical centers across Europe.

We foresee that several centers will cautiously stop the implantation of devices that have not yet been approved as compliant with the MDR. This initial step might pose several challenges especially to small-scale companies, which will be forced to withstand the burden of costs and expertise required to satisfy conditions imposed by the new regulation. In the meantime, it is advisable that surgeons address their requests to manufacturers able to provide their implants in compliance with the MDR.

In parallel, manufacturers will be required to implement all possible solutions to lower the risk of accidents, and this is especially true for high-risk class devices. The Annex I of the MDR explicitly reports that "*manufacturers shall establish, implement, document and maintain a risk management system*" [34], but the regulation does not establish a predefined sequence of tests the prosthesis should undergo before surgical implantation. Intuitively, lack of standardization is intrinsically present in the concept of custom-made devices, therefore each device is completely different and might exhibit peculiarities in its biomechanical features as well as throughout the production process. Therefore, it will be necessary to understand whether every part of the entire process represents a risk or not for the safety of the implant.

By doing so, setting up a bundle of tests to evaluate standardized and not-standardized features will be required. Some of these tests are related to the geometry while others are

feasible on samples such as the evaluation of material properties, porosity, and biological inertness. On the other side, shape-specific assessments will be necessary to investigate all the features that vary according to a patient-specific shape, including support generation analysis and biomechanical FEM evaluation. For instance, considering a TMJ implant, a thin surface will necessarily be subject to greater stress than a thicker one, and might therefore require some adjustments in terms of widening its surface or modifying its topology [35].

However, the regulation also introduces new procedures involving an expert panel, consisting of "advisors appointed by the European Commission on the basis of up-to-date clinical, scientific, or technical expertise in the field", which will be required to scrutinize the clinical evaluation reports for class IIb and class III devices in compliance with a notified body. Additionally, the expert panel will provide guidance for manufacturers in order to comply with the regulation [36]. Communication between clinicians, surgeons, and engineers might be improved by creating laboratories within hospitals where synergistic interaction between different professionals takes place, including discussion of clinical cases, where engineers might learn more concepts on the biology of diseases and the phases of surgery, and design of implants, where conversely surgeons might learn from the experience of engineers and both figures might cooperate to develop highly innovative and effective devices.

The adoption of the novel MDR will introduce profound changes in the field of CMF implants and will deeply affect medical manufacturing. A possible drawback might be the reduction of the number of companies and the variety of devices that today European CMF surgeons can have in their inventory, and in the first instances this might prolong delivery time of devices

However, we expect the benefits will exceed by far such initial limitations. The entry into force of the novel MDR will have a positive impact on broadening our cultural horizons as it will represent a consistent scientific impulse for novel clinical and engineering investigations [37]. Surgeons will potentially be required to draw up new studies on devices compliant with the novel regulations and will necessarily interact with engineers to fulfil all possible vulnerabilities emphasized by the risk analysis. For these reasons, the collaboration between CMF surgeons and engineers in both computational, biomechanical, and clinical investigations concerning implantable devices is going to be critical in the years to come. As a result of clinical investigations, an increasing amount of data will be available on the European database on medical devices (EUDAMED), which will allow a prompt and easy identification of devices to both clinicians and manufacturers, with univocal nomenclature, and will contribute to the development of big data knowledge on medical devices, leading to further investigations and improvements [38].

On the other hand, reviewing our past clinical experience in light of the MDR will emphasize criticalities that were not addressed, introducing an ethical reflection on what we have done before and on responsibility attribution. In summary, we hope that this regulation will provide an even stronger impulse to closely interact with engineers and to educate the surgeons of tomorrow.

## 7. Conclusions

This paper represented the occasion to consolidate a panel of experts around the topic of customized implant creation and regulatory implications. The result of this multidisciplinary interaction is an improved knowledge for surgeons, increasing their awareness in the choice of the correct implant before surgery and allowing decisions to be made not just on clinical bases, but relying upon the understanding of production workflows and possible related errors. We also encourage surgeons to take an active part both in the process of implant development and its post-production assessment, as surgeons are responsible for the choice of the device they plan to implant. It is foreseeable that the MDR will pose initially more barriers to the implantation of devices, and, in particular, customized devices, as well as it will decrease the number of companies satisfying its

requirements; however, we share hope that it will lead to the development of safer devices with inferior complications and improved clinical outcomes.

**Author Contributions:** Conceptualization, methodology, investigation, writing—review and editing, project administration A.T.; software, investigation, writing—review and editing A.B.; formal analysis, data curation and supervision M.S. and G.T.; supervision, L.F. and S.S.; review and editing, E.O.; validation and project administration, M.R. All authors have read and agreed to the published version of the manuscript.

**Funding:** This research received no external funding.

**Institutional Review Board Statement:** Not applicable.

**Informed Consent Statement:** Not applicable.

**Conflicts of Interest:** The authors declare no conflict of interest.

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
