# Peer review of "Current Trends in the Development and Use of Personalized Implants: Engineering Concepts and Regulation Perspectives for the Contemporary Oral and Maxillofacial Surgeon"

_applsci, doi:10.3390/app112411694_

Round 1

Reviewer 1 Report

Constructive communication among engineers, mechanician, scientists, clinician and surgeon are essential for the positive development of surgery and the manufacturers of the related devices. For this goal, many hospitals build up in-house facilities (i.e., small laboratories) which are managed partially or entirely by clinicians, as well as for large-scale companies. However, the regulation of this field is not so easy for the safety and accuracy of the well-targeting patients, particularly for the implant surgery even though it is very necessary. This article aims at reviewing the essential of Medical Device Regulation (MDR) in the regulation the field of implantable devices for the cranio-maxillo-facial skeleton from a multidimensional respective and wants to bridge the gap among clinicians, engineers and surgeons. The authors advanced that it is useful to increase surgeons’ knowledge on production processes related to cranio-maxillo-facial (CMF) implants fabrication and the mechanical, clinical and material engineers shall mutually share their knowledge with surgeons. While engineers should know the urgent concern of surgeon according to the response from their patients. Thus, the company can have some gold suggestion how to design and fabrication their products of higher quality. The surgeon must strictly supervise and actively participate to the design of the implant, while engineers are supportive for technical aspects. Finally, they will lead to the development of safer devices with inferior complications and improved clinical outcomes.

 These ideas are constructive for the policy making of government and the setup of the necessary associate for the produce and service system of surgery. They will favor to the popularization of science to surgeon for enhanced surgery and well-selection of implant devices and confidence to help patients, understand the advantages and limitation of these implant wares. Some attention and double pre-inspection experience before utilization of them and the inspection methods. Provide the detailed information for improvement of these implant ware to the manufacturers scientifically and reasonably. Therefore, this article is very useful for the related industry and the surgeon, as well as the society associate and government, deserved to be published.

There are some suggestions for the authors to consider to improve this article.

  1. Please provide some constructive methods to build up the positive communication among clinicians, engineers and surgeons.
  2. The size and format of Figures and Table are needed to adjust for the journal requirement.
  3. Please double check the grammar and words, such as “To achieve this purpose, a we reunited a task force of experts,” where “a” is one typo.

Author Response

Constructive communication among engineers, mechanician, scientists, clinician and surgeon are essential for the positive development of surgery and the manufacturers of the related devices. For this goal, many hospitals build up in-house facilities (i.e., small laboratories) which are managed partially or entirely by clinicians, as well as for large-scale companies. However, the regulation of this field is not so easy for the safety and accuracy of the well-targeting patients, particularly for the implant surgery even though it is very necessary. This article aims at reviewing the essential of Medical Device Regulation (MDR) in the regulation the field of implantable devices for the cranio-maxillo-facial skeleton from a multidimensional respective and wants to bridge the gap among clinicians, engineers and surgeons. The authors advanced that it is useful to increase surgeons’ knowledge on production processes related to cranio-maxillo-facial (CMF) implants fabrication and the mechanical, clinical and material engineers shall mutually share their knowledge with surgeons. While engineers should know the urgent concern of surgeon according to the response from their patients. Thus, the company can have some gold suggestion how to design and fabrication their products of higher quality. The surgeon must strictly supervise and actively participate to the design of the implant, while engineers are supportive for technical aspects. Finally, they will lead to the development of safer devices with inferior complications and improved clinical outcomes.

 These ideas are constructive for the policy making of government and the setup of the necessary associate for the produce and service system of surgery. They will favor to the popularization of science to surgeon for enhanced surgery and well-selection of implant devices and confidence to help patients, understand the advantages and limitation of these implant wares. Some attention and double pre-inspection experience before utilization of them and the inspection methods. Provide the detailed information for improvement of these implant ware to the manufacturers scientifically and reasonably. Therefore, this article is very useful for the related industry and the surgeon, as well as the society associate and government, deserved to be published.

Dear Reviewer, many thanks for Your honorable appreciation and for sharing with the Authors the aim and the vision of this paper.

There are some suggestions for the authors to consider to improve this article.

  1. Please provide some constructive methods to build up the positive communication among clinicians, engineers and surgeons.

The following sentence was added to Discussion section: “Communication between clinicians, surgeons and engineers might be improved by creating laboratories within hospitals where synergistic interaction between different professionals takes place, including discussion of clinical cases, where engineers might learn more concepts on the biology of diseases and the phases of surgery, and design of implants, where conversely surgeons might learn from the experience of engineers and both figures might cooperate to develop highly innovative and effective devices

  1. The size and format of Figures and Table are needed to adjust for the journal requirement.

Many thanks for this indication. Figures have been separately resubmitted in a single zip archive and at a sufficiently high resolution (300 dpi). Table has been modified according to the journal indications

  1. Please double check the grammar and words, such as “To achieve this purpose, a we reunited a task force of experts,” where “a” is one typo

Many thanks for indicating missed corrections which have been amended in the revised version of the manuscript.

Reviewer 2 Report

Having studied the manuscript of the authors, we can conclude that their research cannot be attributed to scientific works. As the name suggests, this is more populism than scientific research. Indeed, the manuscript is only a "squeeze" from some sources, the number of which is rather limited. The work lacks an analysis of the cited works, does not provide specific data, research results, there is no critical component. All conclusions of the authors are generally known or quite obvious. The manuscript rather resembles an instruction, a chapter of a methodological manual or a text of a report at a conference, but not a scientific work. I would recommend revising the manuscript, shortening the unnecessary text, which is too much for 23 references, and publishing the work as a report abstract.

Author Response

Dear Reviewer,

We accept Your contribution which is an important stimulus to improve the quality of our work. Please, allow us to rewrite the aim of the paper, which might have been misunderstood. As correctly stated, the paper does not present results of a scientific research, nor it presents original data, and subsequently it is not structured as a classic scientific paper, including the canonical “introduction, material and methods, results and discussion”. Similarly, due to the lack of evidence on this topic, it was not possible to present it as a structured systematic review. In addition, companies do not tend to publish their own production parameters, therefore there is no evidence as well on the optimal 3D printing production chain.

In fact, such aims are far beyond the purpose of this paper. Our idea is to write a paper for surgeons, which explores the complexity of additive manufacturing for customized craniofacial implants, a topic that the recently approved MDR has made the center of interest. We desire surgeons to be able to responsibly and consciously evaluate the manufacts they receive from medical companies prior to implantation, and this can only be done by increasing culture on this topic, illustrating to clinicians which are the production chains that generate a personalized craniofacial implant from a CAD model

Taking into high consideration the quality of your response, we consistently improved the scientific backbone of this paper increasing the number of citations for each phase described, as suggested, and providing quantitative data (please, see paragraph 2.1.1 “EBM vs SLM”).

Many sentences have been rephrased to make the text smoothly readable and more intelligible.

Moreover, unnecessary text has been removed (see end of paragraph 5).

Reviewer 3 Report

The paper adresses an very relevant and often overseen problem when working under such interdisciplinary circumstances. I really appreciate the aim to provide engineering perspectives on additve manufacturing for surgeons.

However, from my engineering background, I feel that the presentations on the subject of additive manufacturing, to which i would like to limit my review, are too simple and not sufficiently elaborated.

  • Hardly any quantitative information, only qualitative information provided, see e.g. page 3 line 131 "higher building rate", page 4 line 134 "much lower", and so on...
    Further, references are missing very often here, which would offer the possibility to go into more detail if necessary for the reader.
  • Regarding the listed defects: there already exist proper methods to tackle the different defects, such as geometrical accuracy, for example using geometrical compensation in additve manufacturing.
  • In the paper, I strongly suggest to include the whole process chain for implant manufacturing and not only the additive manufacturing since the subsequent treatment of the part also has major impact on the final quality and characteristics (removing from platform, heat treatments,...).

Please add suitable references to following sections:

  • page 2 line 65 to 83
  • page 3 line 101 to 117
  • page 4
  • page 8
  • page 9
  • page 10 line 319 to page 11 line 356

Additional remarks:

  • page 2 line 81: erase the word "regulation" (already included in abbreviation MDR)
  • page 2 line 88: surgery with lower case "s"
  • page 3 line 99 to 100: please provide a more concise headline
  • page 3 line 117: What is meant by "the fusion itself"?
  • page 4 Figure 1: The text in the figure is not readable and the images have poor quality. The figure caption is meaningless.
  • page 5 Figure 2: The figure caption is meaningless. Please provide more information on the parts shown.
  • page 7/8 Figure 3: poor quality of images, caption on other page.
  • page 11 Figure 4: figure does not contribute to the paper. The caption talks about deformation but actually displacements are shown.

Author Response

The paper adresses an very relevant and often overseen problem when working under such interdisciplinary circumstances. I really appreciate the aim to provide engineering perspectives on additive manufacturing for surgeons.

Dear Reviewer, many thanks for understanding and appreciating our effort to convey to surgeons the conceptual bases of additive manufacturing. The purpose is to aid health professionals to consciously evaluate implants received by a large number of companies. Our belief is that clinicians should have a basic understanding on the processes that lead to the fabrication of a metal implant for cranio-maxillo-facial surgery, including which are possible defects and how they are generated.

However, from my engineering background, I feel that the presentations on the subject of additive manufacturing, to which i would like to limit my review, are too simple and not sufficiently elaborated.

Although the paper is addressed to surgeons and was developed with the help of our engineering team, we took seriously into account Your valuable suggestion, therefore we requested a more detailed contribution from engineers and definitely expanded our description on additive manufacturing processes.

  • Hardly any quantitative information, only qualitative information provided, see e.g. page 3 line 131 "higher building rate", page 4 line 134 "much lower", and so on...
    Further, references are missing very often here, which would offer the possibility to go into more detail if necessary for the reader.

We fully agree with Your considerations and we modified the paper accordingly. At the end of page 3, you can find the following paragraphs improved with quantitative information and specific references:
EBM technology has usually a higher building rate in respect to the SLM, therefore it is more suitable for the production of complex shapes, for example lattice structures; EBM technology has usually a higher building rate compared to SLM. This is due to two main factors: first, layer thickness has higher values (50 – 100µm in respect to the 20-100µm of the SLM process); secondly, the immediate electron beam motion from one location to another, thanks to the instantaneous response of magnetic coils, can considerably speed up component fabrication (for example, electron beam speed can reach 8000m/s in ARCAM machines[9], with the laser reach 7 m/s in EOS machines[10]). For instance, in the production of truss-like structures (lattice structures), EBM is to be chosen”,

and also “the surface finishing of a component produced with EBM is much lower in respect to an SLM product. Is it sufficient to consider that the roughness parameters of EBM "as built" specimens are about twice those of SLM "as built" ones: in fact the roughness of EBM is 30 to 40 um while for SLM specimens is 11 to 18 um. This is mainly due to the fact that the layer thickness, powder size and melting pool size in the case of SLM are half the one used in EBM specimens [11]”;

as well as “the components produced with SLM undergo strong thermic gradients during the 3D printing process, since the preheating temperature of the powders is generally low, since the temperature of the chamber in SLM process is the enviroment temperature, assumed to be 293K[12]. That is why they need some thermic post treatments in order to reduce the residual stresses that take origin inside the material because of the rapid and iterative phase change of the metal (solid-liquid-solid). On the contrary, in the EBM technology the temperature of the powder bed is higher and ensures the absence of thermal stresses inside the melted material, around 870K during the melting process[13]”

  • Regarding the listed defects: there already exist proper methods to tackle the different defects, such as geometrical accuracy, for example using geometrical compensation in additve manufacturing.

In our intention, the paper aims to be a sort of “handbook for the clinician”, providing surgeons requesting customized implants the knowledge to consciously perform a visual inspection once they receive the implant before surgery. The aim is that surgeons are able to understand by themselves whether the implant shows good quality or not, and this is especially important for those hospitals where engineers are not present. The correct visual inspection of the implant, as our experience has shown to us, can prevent a dangerous surgery where an unsuitable device is positioned, with problems of correct fitting on anatomical surfaces or other defects.

Moreover, as surgeons are being progressively more involved also in the design phase of the implants they require, it is important that they keep in mind some advices that come from engineering expertise.

We obviously do not desire to teach companies how to evaluate their products, since we believe that specialized companies implement completely different and more specific methods in the production chain.

  • In the paper, I strongly suggest to include the whole process chain for implant manufacturing and not only the additive manufacturing since the subsequent treatment of the part also has major impact on the final quality and characteristics (removing from platform, heat treatments,...).

Similarly, since this paper is addressed to surgeons, we focused on the process chain of SLM and EBM. We valued Your respected request and added more information under section 2.1.1. “EBM vs SLM”; moreover, a consistent description was added on the topic of support structures. In surgical planning, the choice of regions bearing support structures has a major impact on the surgery itself, as supports should not be placed along surfaces in contact with bone, not to alter the precise fitting of the implant.

Please add suitable references to following sections:

  • page 2 line 65 to 83

the following citations have been added:

Lee, D.H.; Reasoner, K.; Stewart, A. From Concept to Counter: A Review of Bringing an Orthopaedic Implant to Market. Journal of the American Academy of Orthopaedic Surgeons 2020, 28, e604–e611, doi:10.5435/JAAOS-D-20-00017.

Aimar, A.; Palermo, A.; Innocenti, B. The Role of 3D Printing in Medical Applications: A State of the Art. Journal of Healthcare Engineering 2019, 2019, 1–10, doi:10.1155/2019/5340616.

  • page 3 line 101 to 117

Gibson, I.; Rosen, D.W.; Stucker, B. Additive Manufacturing Technologies: 3D Printing, Rapid Prototyping, and Direct Digital Manufacturing; 2. ed.; Springer: New York, NY, 2015; ISBN 978-1-4939-2113-3.

page 4

Arcam, A.B. Arcam Q20 Technical Data. 2015. Available online: www.arcam.com/wp-content/uploads/Arcam-Q20-final.pdf.

EOS. Laser sintering system EOSINT M 280 for the production of tooling inserts, prototype parts and end products directly in metal. In The Technology: Laser Sintering—The Key to e-Manufacturing; EOS GmbH Electro Optical Systems Corporate Headquarters: Krailling, Germany; Munich, Germany, 2017

Vayssette, B.; Saintier, N.; Brugger, C.; El May, M. Surface Roughness Effect of SLM and EBM Ti-6Al-4V on Multiaxial High Cycle Fatigue. Theoretical and Applied Fracture Mechanics 2020, 108, 102581, doi:10.1016/j.tafmec.2020.102581.

Ansari, M.J.; Nguyen, D.-S.; Park, H.S. Investigation of SLM Process in Terms of Temperature Distribution and Melting Pool Size: Modeling and Experimental Approaches. Materials 2019, 12, 1272, doi:10.3390/ma12081272.

Gokuldoss, P.K.; Kolla, S.; Eckert, J. Additive Manufacturing Processes: Selective Laser Melting, Electron Beam Melting and Binder Jetting—Selection Guidelines. Materials 2017, 10, 672, doi:10.3390/ma10060672.

Zhao, X.; Li, S.; Zhang, M.; Liu, Y.; Sercombe, T.B.; Wang, S.; Hao, Y.; Yang, R.; Murr, L.E. Comparison of the Microstructures and Mechanical Properties of Ti–6Al–4V Fabricated by Selective Laser Melting and Electron Beam Melting. Materials & Design 2016, 95, 21–31, doi:10.1016/j.matdes.2015.12.135.

Koike, M.; Greer, P.; Owen, K.; Lilly, G.; Murr, L.E.; Gaytan, S.M.; Martinez, E.; Okabe, T. Evaluation of Titanium Alloys Fabricated Using Rapid Prototyping Technologies—Electron Beam Melting and Laser Beam Melting. Materials 2011, 4, 1776–1792, doi:10.3390/ma4101776.

Jiang, J.; Xu, X.; Stringer, J. Support Structures for Additive Manufacturing: A Review. Journal of Manufacturing and Materials Processing 2018, 2, 64, doi:10.3390/jmmp2040064.

  • page 8

Rucka, M. Special Issue: “Non-Destructive Testing of Structures.” Materials 2020, 13, 4996, doi:10.3390/ma13214996.

Turó, A.; Chávez, J.A.; García-Hernández, M.J.; Bulkai, A.; Tomek, P.; Tóth, G.; Gironés, A.; Salazar, J. Ultrasonic Inspection System for Powder Metallurgy Parts. Measurement 2013, 46, 1101–1108, doi:10.1016/j.measurement.2012.10.016.

Millon, C.; Vanhoye, A.; Obaton, A.-F.; Penot, J.-D. Development of Laser Ultrasonics Inspection for Online Monitoring of Additive Manufacturing. Welding in the World 2018, 62, 653–661, doi:10.1007/s40194-018-0567-9.

Taud, H.; Martinez-Angeles, R.; Parrot, J.F.; Hernandez-Escobedo, L. Porosity Estimation Method by X-Ray Computed Tomography. Journal of Petroleum Science and Engineering 2005, 47, 209–217, doi:10.1016/j.petrol.2005.03.009.

Farber, L.; Tardos, G.; Michaels, J.N. Use of X-Ray Tomography to Study the Porosity and Morphology of Granules. Powder Technology 2003, 132, 57–63, doi:10.1016/S0032-5910(03)00043-3.

  • page 9

Additive Manufacturing Handbook: Product Development for the Defense Industry; Badiru, A.B., Valencia, V.V., Liu, D., Eds.; 1st ed.; CRC Press, 2017; ISBN 978-1-315-11910-6.

  • page 10 line 319 to page 11 line 356

regulations have explititely been stated, and include ISO 10993:2021, ISO 11737, ISO 17025, ISO 14801:2007, ISO 16428:2005, ISO 17853:2011

Additional remarks:

  • page 2 line 81: erase the word "regulation" (already included in abbreviation MDR)

We did as required

  • page 2 line 88: surgery with lower case "s"

We modified accordingly

  • page 3 line 99 to 100: please provide a more concise headline

Now it is: Additive manufacturing concepts for the CMF surgeon

  • page 3 line 117: What is meant by "the fusion itself"?

This sentence has been rephrased into: However, the clinician should understand certain features of the fusion process and some general principles of defects formation related to this technology.

  • page 4 Figure 1: The text in the figure is not readable and the images have poor quality. The figure caption is meaningless.

The caption has been rephrased into: EBM and SLM technology principles: (a) EBM; (b) SLM. A high resolution version of the image has been uploaded in the text and the whole image has been made larger.

  • page 5 Figure 2: The figure caption is meaningless. Please provide more information on the parts shown.

The figure caption has been rephrased into: “3D printed implants manufactured by using SLM and used for surgeries at our institution include customized reconstruction plates, temporomandibular joint prostheses and full mandibular prostheses used for reconstructive purposes in oncologic patients”.

  • page 7/8 Figure 3: poor quality of images, caption on other page.

The figure has been replaced with a high resolution replica and has been widened. The caption now lies within the same page.

  • page 11 Figure 4: figure does not contribute to the paper. The caption talks about deformation but actually displacements are shown.

The figure might be useful to surgeons to understand the concept of finite element analysis. The caption has been made more precise, as pointed out: Finite element analysis performed on the final design reveals the total deflection of the  component under the load of a cantilever force whose intensity is far beyond the physiological working conditions. The maximum calculated deflection together with the maximum stress analysis reveals if the component is suitable for implant or not”.

Round 2

Reviewer 2 Report

After reading the revised manuscript, it is worth noting that the authors have made significant adjustments to the text. Nevertheless, the manuscript requires significant revision.

Title:

The word "era" in the title sounds like populism and is not objective. MDR 2022 may appear soon. Can this be called an era?

In addition, scientific work should not be guided by normative documentation. Requirements are created in relation to existing materials and procedures only. But this does not stop research in the field of new materials. There will be new materials - there will be new requirements for them, and not vice versa.

The title contains the term "learn". As noted earlier, the manuscript resembles a study guide. Admonitions are also found throughout the text, which will be discussed later. However, scientific work still has a different purpose.

The use of abbreviations in the name is undesirable. In general, it is better to change the name, for example "Current trends in the development and use of individual implants."

At the same time, the entire importance of interdisciplinary research can be reflected in the abstract. The abstract should also indicate the main results of the authors' research and remove all unnecessary.

Introduction

In the introduction, more legal, legislative and economic aspects are displayed, which in no way correlate with the tasks of materials science. If the purpose of the manuscript is to familiarize readers with European legislation, then the work should be published in more specialized literature. From a scientific point of view, the introduction should briefly reflect the current state of science in the field of research: what has been developed today, what are the achievements and shortcomings in the field of surgery and materials science, as well as the points of contact of these disciplines.

Section 2.1

The section is written as an instruction, many recommendations are given, and unnecessary explanations are given. At the same time, no confirmation or scientific justification of the information provided is provided. Do you need this section?

Section 2.3

The section is a set of recommendations, not a scientific text. It should be carefully reworked. You can specify which specific requirements products must meet, if possible in numerical form, scientifically explain why they must meet these requirements and how this can be achieved. You can indicate the advantages and disadvantages of a particular manipulation.

Sections 2-4

I believe that the information presented in this section will be too difficult for doctors to perceive. Therefore, when naming sections, one should not strictly focus on any specialists, but should be limited to general phrases. You should also focus more not on the regulations, but on the result and perspective.

Section 5.

The section does not contain scientifically significant information, it should be deleted. Some of the information from it can be transferred to the introduction and conclusion.

Section 6

This section contains only recommendations. These recommendations are important for teaching, but not important for science.

Conclusion.

The section should be rewritten to indicate what conclusions the authors came to as a result of their research. What are the advantages of modern materials? What do surgeons need? How can and should they unite their efforts?

Author Response

After reading the revised manuscript, it is worth noting that the authors have made significant adjustments to the text. Nevertheless, the manuscript requires significant revision.

Dear Reviewer,

Many thanks for reconsidering our paper and taking into account improvements made. We are grateful for Your further suggestions and we are committed to follow them to further increase the quality of this paper.

Title:

The word "era" in the title sounds like populism and is not objective. MDR 2022 may appear soon. Can this be called an era?

Many thanks for providing an interesting perspective to reflect on. Indeed, we agree and therefore we provided a more objective title: “Current trends in the development and use of personalized implants: engineering concepts and regulation perspectives for the contemporary oral and maxillofacial surgeon

In addition, scientific work should not be guided by normative documentation. Requirements are created in relation to existing materials and procedures only. But this does not stop research in the field of new materials. There will be new materials - there will be new requirements for them, and not vice versa.

We absolutely agree with these considerations. Science and progress are the factors that drive innovation and only subsequently laws are promulged to provide a guidance and a set of rules. We believe that customization of craniofacial implants represents the perfect example, because surgical innovation has improved the life of many patients even before the entry into force of MDR 2021. However, the regulation is expected to improve such processes, providing standardization and quality control.

The title contains the term "learn". As noted earlier, the manuscript resembles a study guide. Admonitions are also found throughout the text, which will be discussed later. However, scientific work still has a different purpose.
The use of abbreviations in the name is undesirable. In general, it is better to change the name, for example "Current trends in the development and use of individual implants."

Your considerations are valuable and supportive for our manuscript. Moreover, we appreciate Your contribution to the title of the manuscript, which we implemented. Indeed, the term “learn” is misleading. Therefore, taking into account Your suggestions, the final title will be: Current trends in the development and use of personalized implants: engineering concepts and regulation perspectives for the contemporary oral and maxillofacial surgeon

At the same time, the entire importance of interdisciplinary research can be reflected in the abstract. The abstract should also indicate the main results of the authors' research and remove all unnecessary.

We agree with this valuable consideration, therefore we removed the unnecessary previous final sentence, and replaced it with the following, emphasizing the importance of interdisciplinary research and its results: “The result of interdisciplinary research is a paper aiming to provide surgeons with the knowledge on the fundamental processes of additive manufacturing, increasing the clinician’s awareness on the evaluation of a customized implant before surgery and on the underlying regulatory framework

Introduction

In the introduction, more legal, legislative and economic aspects are displayed, which in no way correlate with the tasks of materials science. If the purpose of the manuscript is to familiarize readers with European legislation, then the work should be published in more specialized literature. From a scientific point of view, the introduction should briefly reflect the current state of science in the field of research: what has been developed today, what are the achievements and shortcomings in the field of surgery and materials science, as well as the points of contact of these disciplines.

Many thanks for pointing this out. Introduction section has been extensively rewritten, and most aspects related to legislative aspects, including ISO standards, have been deleted, as they were too specific and not directly related to material science and manufacturing technology. The only obvious reference to MDR was left, as a general example to illustrate the importance of a regulatory framework. Moreover, some parts of the introduction have been rephrased to guide the reader to the main focus of this paper, that is knowledge of manufacturing processes, that generally represent a “blind phase” for the surgeon: “Implantable devices deserve a separate discussion since they are produced by external companies using additive manufacturing technologies. Implants are ordered and received ready for surgery, and oftentimes no validation of their quality is performed by surgeons, that generally have little or no knowledge on production processes related to cranio-maxillo-facial (CMF) implants fabrication”.
With such improvements, the main purpose of this paper remains clear: “The aim of this paper is to illustrate the complexity of the topic concerning patient-specific implant production from a multidimensional point of view, especially in relation to manufacturing technologies and final product evaluation, with the hope to provide surgeons with the indispensable knowledge to responsibly evaluate a customized craniofacial implant before surgery, as well as to better define related regulatory aspects”.

Section 2.1

The section is written as an instruction, many recommendations are given, and unnecessary explanations are given. At the same time, no confirmation or scientific justification of the information provided is provided. Do you need this section?

Many thanks for this suggestion. We removed sentences resembling “an instruction”, including: “it is important for the CMF surgeon to understand some principles…” and “However, the clinician should understand certain features of the fusion process…”. Nevertheless, we would be grateful if You could allow us to maintain certain introductory sentences that allow to better introduce the concept of additive manufacturing for surgeons and its importance in the creation of customized devices.

Section 2.3

The section is a set of recommendations, not a scientific text. It should be carefully reworked. You can specify which specific requirements products must meet, if possible in numerical form, scientifically explain why they must meet these requirements and how this can be achieved. You can indicate the advantages and disadvantages of a particular manipulation.

All the recommendations derive from our experience of intensive activity of 3D printing with SLM technology performed within the LAMAFVG laboratory associated with the Engineering Department of the University of Udine (DPIA), using the SLM machine LaserCUSING M2 by Concept Laser. Together with everyday case studies, we furthermore developed a benchmark containing several critical features and we inspected it using a 3D laser scanner (Absolute Arm by Hexagon that reaches a precision of ±0.01mm) to understand the geometrical inaccuracies of this kind of technology depending on several different parameters such as thickness, inclination, length of bridge-like structures, grid density, surface quality.

As these explanations are too detailed and not suitable for a paper destined to surgeons, we prefer to omit them. However, given Your appreciated request, we decided to provide You with a detailed response.

Moreover, we attach a figure illustrating a benchmark developed at the LAMA FVG Laboratory associated with the University of Udine for assessing SLM technological limits in terms of geometrical features feasibility, dimensional accuracy and repeatability, surface quality. (a), (b) and (c) CAD model; (d) and (e) real 3D printed part visual inspection; (f) dimensional inspection through laser scanner. Parameters and recommendations given throughout the manuscript refer to this set of analyses.

Sections 2-4

I believe that the information presented in this section will be too difficult for doctors to perceive. Therefore, when naming sections, one should not strictly focus on any specialists, but should be limited to general phrases. You should also focus more not on the regulations, but on the result and perspective.

As stated before, this paper necessarily contains some technical information addressed to surgeons, who should seek for engineers’ consult to interpret them. Thus the paper has the purpose to encourage the creation of multidisciplinary study groups. Since this paper was written by a group composed of maxillofacial surgeons, we had to mention the specialty and field in which such personalized implants are used. Of course, the general conclusions of this paper are true also for other specialists adopting custom made implants, such as orthopedists, spine surgeons and so on. Results and perspectives have been better delineated in the revised version of Introduction and Conclusion sections, as You suggested.

Section 5.

The section does not contain scientifically significant information, it should be deleted. Some of the information from it can be transferred to the introduction and conclusion.

We understand and respect Your point of view, therefore we shortened some parts by deleting less objective sentences, for instance: “software, which is not strictly encased in class I or IIa, but a novel, fluid classification is introduced…”. This sentence was a reflection made by the Authors based on the MDR, but it was removed as it was not scientifically significant.
Nevertheless, we would be grateful if You could allow us to maintain this shortened version of section 5, as point-of-care 3D printing labs are every day more present in modern hospitals, and a significant part of the surgical planning and production of non-implantable devices is performed in such laboratories (see for instance: Williams FC, Hammer DA, Wentland TR, Kim RY. Immediate Teeth in Fibulas: Planning and Digital Workflow With Point-of-Care 3D Printing. J Oral Maxillofac Surg. 2020 Aug;78(8):1320-1327. doi: 10.1016/j.joms.2020.04.006; Abo Sharkh H, Makhoul N. In-House Surgeon-Led Virtual Surgical Planning for Maxillofacial Reconstruction. J Oral Maxillofac Surg. 2020 Apr;78(4):651-660. doi: 10.1016/j.joms.2019.11.013; Bekisz JM, Liss HA, Maliha SG, Witek L, Coelho PG, Flores RL. In-House Manufacture of Sterilizable, Scaled, Patient-Specific 3D-Printed Models for Rhinoplasty. Aesthet Surg J. 2019 Feb 15;39(3):254-263. doi: 10.1093/asj/sjy158”.
In fact, one of the main purposes of our paper is to raise attention on the processes that involve the creation of personalized devices, of which, as mentioned in introduction section, additively manufactured metal implants are only a part, but a consistent amount of devices is also represented by other surgical tools, including guides for osteotomies and implant positioning, molds to create cranial plates, templates for orbital surgery and so on.

Section 6

This section contains only recommendations. These recommendations are important for teaching, but not important for science.

Many thanks for this indication. Sentences containing recommendations, which often contained the terms “should” or “advise”, have been deleted from Section 6, for instance: “as a consequence, surgeons should understand some engineering principles related to defect avoidance…” and “we strongly advise that such figures are present even in hospitals and universities…”

Conclusion.

The section should be rewritten to indicate what conclusions the authors came to as a result of their research. What are the advantages of modern materials? What do surgeons need? How can and should they unite their efforts?

Conclusion section has been extensively rewritten specifying the need for surgeons to increase knowledge on production worklows and underlying the importance of multidisciplinary interaction.

In summary, a substantial revision of the paper was performed by all Authors, and we are thankful for the suggestions provided.

Reviewer 3 Report

The paper was overworked pretty well and I would suggest to accept in present form.

Author Response

Dear Reviewer, many thanks for Your help and advices to improve our manuscript. We are pleased of Your appreciation.

Kind regards

The Authors

Round 3

Reviewer 2 Report

The authors made changes to the manuscript that improved its quality. I believe that the manuscript can be accepted by publication.